# Removing the Feature Correlation Effect of Multiplicative Noise

**Zijun Zhang**
University of Calgary
zijun.zhang@ucalgary.ca

**Yining Zhang**
University of Calgary
yining.zhang1@ucalgary.ca

**Zongpeng Li**
Wuhan University
zongpeng@whu.edu.cn

## Abstract

Multiplicative noise, including dropout, is widely used to regularize deep neural networks (DNNs), and is shown to be effective in a wide range of architectures and tasks. From an information perspective, we consider injecting multiplicative noise into a DNN as training the network to solve the task with noisy information pathways, which leads to the observation that multiplicative noise tends to increase the correlation between features, so as to increase the signal-to-noise ratio of information pathways. However, high feature correlation is undesirable, as it increases redundancy in representations. In this work, we propose non-correlating multiplicative noise (NCMN), which exploits batch normalization to remove the correlation effect in a simple yet effective way. We show that NCMN significantly improves the performance of standard multiplicative noise on image classification tasks, providing a better alternative to dropout for batch-normalized networks. Additionally, we present a unified view of NCMN and shake-shake regularization, which explains the performance gain of the latter.

## 1 Introduction

State-of-the-art deep neural networks are often over-parameterized to deliver more expressive power. For instance, a typical convolutional neural network (CNN) for image classification can consist of tens to hundreds of layers, and millions to tens of millions of learnable parameters [1, 2]. To combat overfitting, a variety of regularization techniques have been developed. Examples include dropout [3], DropConnect [4], and the recently proposed shake-shake regularization [5]. Among them, dropout is arguably the most popular, due to its simplicity and effectiveness in a wide range of architectures and tasks, e.g., convolutional neural networks (CNNs) for image recognition [6], and recurrent neural networks (RNNs) for natural language processing (NLP) [7].

Nonetheless, we observe a side effect of dropout that has long been ignored. That is, it tends to increase the correlation between the features it is applied to, which reduces the efficiency of representations. It is also known that decorrelated features can lead to better generalization [8, 9, 10, 11]. Thus, this side effect may counteract, to some extent, the regularization effect of dropout.

In this work, we demonstrate the feature correlation effect of dropout, as well as other types of multiplicative noise, through analysis and experiments. Our analysis is based on a simple assumption that, in order to reduce the interference of noise, the training process will try to maximize the *signal-to-noise ratio* (SNR) of representations. We show that the tendency of increasing the SNR will increase feature correlation as a result. To remove the correlation effect, it is possible to resort to feature decorrelation techniques. However, existing techniques penalize high correlation explicitly; they either introduce a substantial computational overhead [10], or yield marginal improvements [11]. Moreover, these techniques require extra hyperparameters to control the strength of the penalty, which further hinders their practical application.

We propose a simple yet effective approach to solve this problem. Specifically, we first decompose noisy features into the sum of two components, a signal component and a noise component, and then truncate the gradient through the latter, i.e., treat it as a constant. However, naively modifying the gradient would encourage the magnitude of features to grow in order to increase the SNR, causing optimization difficulties. We solve this problem by combining the aforementioned technique with batch normalization [12], which effectively counteracts the tendency of increasing feature magnitude. The resulting method, non-correlating multiplicative noise (NCMN), is able to reduce the correlation between features, reaching a level even lower than that without multiplicative noise. More importantly, it significantly improves the performance of standard multiplicative noise on image classification tasks.

As another contribution of this work, we further investigate the connection between NCMN and shake-shake regularization. Despite its impressive performance, how shake-shake works remains elusive. We show that the noise produced by shake-shake has a similar form to that of a NCMN variant, and both NCMN and shake-shake achieve superior generalization performance by avoiding feature correlation.

The rest of this paper is organized as follows. In Section 2, we define a general form of multiplicative noise, and identify the feature correlation effect. In Section 3, we first propose NCMN, which we show through analysis is able to remove the feature correlation effect; then we develop multiple variants of NCMN, and provide a unified view of NCMN and shake-shake regularization. In Section 4, we provide empirical evidence of our analysis, and evaluate the performance of the proposed methods.[1]

## 2 Motivation

### 2.1 Multiplicative Noise

Let $x_i$ be the activation of hidden unit $i$, we consider multiplicative noise, $u_i$, which is applied to the activations of layer $l$ as

$$\tilde{x}_i = u_i x_i, \forall i \in \mathcal{H}^l. \tag{1}$$

Here, $\mathcal{H}^l$ represents the set of hidden units in layer $l$. For simplicity, we restrict our analysis to fully-connected layers without batch normalization, and will later extend it to batch-normalized layers and convolutional layers. When used for regularization purpose, multiplicative noise is typically applied at training time, and removed at test time. Consequently, the noise should satisfy $\mathbb{E}[u_i] = 1$, such that $\mathbb{E}[\tilde{x}_i] = x_i$.

The noise mask, $u_i$, can be sampled from various distributions, as exemplified by Bernoulli, Gaussian, and uniform distributions. We take dropout and DropConnect as examples. For dropout, let $m_i$ be the dropout mask sampled from a Bernoulli distribution, $\text{Bern}(p)$, then the equivalent multiplicative noise is given by $u_i = m_i/p$. DropConnect is slightly different from dropout, in that the dropout mask is independently sampled for each weight instead of each activation. Thus, we denote the dropout mask by $m_{ij}$, where $j \in \mathcal{H}^{l+1}$, then we have $u_{ij} = m_{ij}/p$ and

$$\tilde{x}_{ij} = u_{ij} x_i, \forall i \in \mathcal{H}^l, j \in \mathcal{H}^{l+1}. \tag{2}$$

Comparing Eq. (2) with Eq. (1), we observe that applying multiplicative noise to weights is equivalent to applying it to activations, except that the noise mask is independently sampled for each hidden unit in the upper layer. Therefore, without loss of generality, we consider multiplicative noise of the form in Eq. (1) in the following discussion.

Compared to other types of noise, such as additive isotropic noise, multiplicative noise can adapt the scale of noise to the scale of features, which may contribute to its empirical success.

### 2.2 The Feature Correlation Effect

As a regularization technique, dropout, and other multiplicative noise, improve generalization by preventing feature co-adaptation [3]. From an information perspective, injecting noise into a neural network can be seen as training the model to solve the task with *noisy information pathways*. To better solve the task, a simple and natural strategy that can be learned is to increase the signal-to-noise ratio (SNR) of the information pathways.

Concretely, the noisy activations of layer $l$ is aggregated by a weighted sum to form the pre-activations (without biases) of layer $l + 1$ as

$$z_j = \sum_{i \in \mathcal{H}^l} w_{ij} \tilde{x}_i, \forall j \in \mathcal{H}^{l+1}, \tag{3}$$

where $w_{ij}$ is the weight between unit $i$ in layer $l$ and unit $j$ in layer $l + 1$. Although we cannot increase the SNR of $\tilde{x}_i$ due to the multiplicative nature of the noise, it is possible to increase the SNR of $z_j$ instead. In the following, we omit the range of summation when it is over $\mathcal{H}^l$.

We now focus on the pre-activation, $z_j$, of an arbitrary unit in layer $l + 1$, and define its signal and noise components, respectively, as

$$z_j^s = \sum_i w_{ij} x_i, \text{ and } z_j^n = \sum_i w_{ij} v_i x_i, \tag{4}$$

where $v_i = u_i - 1$, such that $\mathbb{E}[v_i] = 0$, and $z_j = z_j^s + z_j^n$. Then we can model the tendency of increasing the SNR of information pathways by the following implicit objective function:

$$\text{maximize} \quad \text{SNR}(z_j) = \frac{\mathbb{E}\left[\left(z_j^s - \mathbb{E}[z_j^s]\right)^2\right]}{\mathbb{E}\left[\left(z_j^n\right)^2\right]}. \tag{5}$$

Here, the expectations are taken with respect to both $x_i$ and $v_i$, $\forall i \in \mathcal{H}^l$. Note that in Eq. (5), we subtract the constant component, $\mathbb{E}[z_j^s]$, from the signal, since it does not capture the variation of input samples. Let $\sigma^2 = \text{Var}[v_i], \forall i \in \mathcal{H}^l$, we have

$$\text{SNR}(z_j) = \frac{1}{\sigma^2}\left[1 + \frac{2\mathbb{E}\left[\sum_{i' \neq i}\sum_i (w_{ij}x_i)(w_{i'j}x_{i'})\right] - \mathbb{E}[z_j^s]^2}{\mathbb{E}\left[\sum_i (w_{ij}x_i)^2\right]}\right]. \tag{6}$$

Thus, the objective function in Eq. (5) is equivalent to the following:

$$\text{maximize} \quad \frac{2\mathbb{E}\left[\sum_{i' \neq i}\sum_i (w_{ij}x_i)(w_{i'j}x_{i'})\right]}{\mathbb{E}\left[\sum_i (w_{ij}x_i)^2\right]} - \frac{\mathbb{E}[z_j^s]^2}{\mathbb{E}\left[\sum_i (w_{ij}x_i)^2\right]}. \tag{7}$$

Intuitively, the first term in Eq. (7) tries to maximize the correlation between each pair of $w_{ij}x_i$ and $w_{i'j}x_i$, where $i \neq i'$. Although it is not the same as the commonly used Pearson correlation coefficient, it can be regarded as a generalization of the latter to multiple variables. Since $w_{ij}$ can be either positive or negative, maximizing the correlations between $w_{ij}x_i$'s essentially increases the magnitudes of the correlations between $x_i$'s, and hence causing the feature correlation effect. The second term in Eq. (7) penalizes non-zero values of $\mathbb{E}[z_j^s]$, and does not affect feature correlations.

For batch-normalized layers, if batch normalization is applied to $z_j$ (as a common practice), the numerator and denominator of Eq. (5) will be divided by the same factor, $\sqrt{\text{Var}[z_j]}$. Therefore, Eq. (5) remains the same, and the analysis still holds.

For convolutional layers, one should consider $\mathcal{H}^{l+1}$ as the set of convolutional kernels or feature maps in layer $l + 1$, and $\mathcal{H}^l$ as the set of input activations at each spatial location. Accordingly, the inputs at different spatial locations are considered as different input samples. Since adjacent activations in the same feature map are often highly correlated, sharing the same noise mask across different spatial locations is shown to be more effective [13]. In this setting, the feature correlation effect tends to be more prominent between activations in different feature maps than that in the same feature map.

## 3 Methods

### 3.1 Non-Correlating Multiplicative Noise

A high correlation between features increases redundancy in neural representations, and can thus reduce the expressive power of neural networks. To remove this effect, one can directly penalize the correlation between features as part of the objective function [10], which, however, introduces a substantial computational overhead. Alternatively, one can penalize the correlation between the weight vectors (or convolutional kernels) of different hidden units [11], which is less computationally

expensive, but yields only marginal improvements. Moreover, both approaches require manually-tuned penalty strength. A more desirable approach is to simply avoid feature correlation in the first place, rather than counteracting it with other regularization techniques.

From Eq. (5) we observe that, if we consider the denominator $\mathbb{E}\left[\left(z_j^n\right)^2\right]$ as a constant, i.e., ignore its gradient during training, the objective function is equivalent to

$$\text{maximize} \quad \mathbb{E}\left[\left(z_j^s - \mathbb{E}\left[z_j^s\right]\right)^2\right]. \tag{8}$$

Eq. (8) implies that if we ignore the gradient of the noise component, $z_j^n$, the tendency of increasing the SNR of $z_j$ will attempt to increase the variance of the signal component, $z_j^s$, instead of increasing the feature correlation of the lower layer. However, we find in practice that such modification to the gradient causes optimization difficulties, preventing the training process from converging. Fortunately, by using batch normalization, the remedy for this problem is surprisingly simple.

Concretely, we apply batch normalization to $z_j$ as

$$\hat{z}_j = \text{BN}\left(z_j; z_j^s\right) = \frac{z_j - \mathbb{E}\left[z_j^s\right]}{\sqrt{\text{Var}\left[z_j^s\right]}}. \tag{9}$$

We neglect the small difference between the true mean/variance, and the sample mean/variance, and adopt the same notation for simplicity. Note that in Eq. (9), $z_j$ is normalized using the statistics of $z_j^s$, which is slightly different from standard batch normalization. We now consider the SNR of the new pre-activation, $\hat{z}_j$. The signal and noise components of $\hat{z}_j$ are respectively

$$\hat{z}_j^s = \frac{z_j^s - \mathbb{E}\left[z_j^s\right]}{\sqrt{\text{Var}\left[z_j^s\right]}}, \text{ and } \hat{z}_j^n = \hat{z}_j - \hat{z}_j^s = \frac{z_j^n}{\sqrt{\text{Var}\left[z_j^s\right]}}. \tag{10}$$

For clarity, we define an *identity* function, $\text{AsConst}\left(\cdot\right)$, meaning that the argument of the function is considered as a constant during the backpropagation phase, or in other words, its gradient is set to zero by the function. We then substitute $\hat{z}_j$ with

$$\hat{z}_j' = \hat{z}_j^s + \text{AsConst}\left(\hat{z}_j^n\right) = \text{BN}\left(z_j^s\right) + \text{AsConst}\left(\text{BN}\left(z_j; z_j^s\right) - \text{BN}\left(z_j^s\right)\right), \tag{11}$$

such that the noise component is considered as a constant. Therefore, maximizing $\text{SNR}\left(\hat{z}_j'\right)$ is equivalent to

$$\text{maximize} \quad \mathbb{E}\left[\left(\hat{z}_j^s\right)^2\right]. \tag{12}$$

Due to the use of batch normalization, Eq. (12) is a constant with respect to each sample of $z_j^s$, and thus we have $\partial \mathbb{E}\left[\left(\hat{z}_j^s\right)^2\right]/\partial z_j^s\left(m\right) = 0, \forall m \in \mathcal{B}$, where $\mathcal{B}$ denotes a set of mini-batch samples, and $z_j^s\left(m\right)$ denotes the value of $z_j^s$ corresponding to sample $m$. Therefore, we can now remove the feature correlation effect of multiplicative noise without causing optimization difficulties. We refer to this approach as non-correlating multiplicative noise (NCMN).

We also note that the non-standard batch normalization used in Eq. (9) is not necessary in practice. To take advantage of existing optimized implementations of batch normalization, we can modify Eq. (11) as follows:

$$\hat{z}_j' = \text{BN}\left(z_j^s\right) + \text{AsConst}\left(\text{BN}\left(z_j\right) - \text{BN}\left(z_j^s\right)\right). \tag{13}$$

In this case, to keep the forward pass consistent between training and testing, the running mean and variance should be calculated based on $z_j$, rather than $z_j^s$.

Interestingly, Eq. (11) and Eq. (13) can be seen as adding a noise component to batch-normalized $z_j^s$, where the noise is generated in a special way and passed through the $\text{AsConst}\left(\cdot\right)$ function. However, the analysis in this section does not depend on the particular choice of the noise, as long as it is considered as a constant. This observation leads to a unified view of multiple variants of NCMN and shake-shake regularization, as discussed in the following section.

## 3.2 A Unified View of NCMN and Shake-shake Regularization

In Eq. (11), the noise component, $\hat{z}_j^n$, is generated indirectly from the multiplicative noise applied to the lower layer activations. Assuming the independence of $v_i$'s, we have $\mathbb{E}\left[\hat{z}_j^n\right] = 0$, and

$$\text{Var}\left[\hat{z}_j^n\right] = \frac{\sigma^2}{\text{Var}\left[z_j^s\right]} \sum_i \left(w_{ij} x_i\right)^2 = \frac{\sigma^2}{\text{Var}\left[z_j^s\right]} \left[\left(z_j^s\right)^2 - 2\sum_{i' \neq i}\sum_i \left(w_{ij} x_i\right)\left(w_{i'j} x_{i'}\right)\right], \tag{14}$$

which implies that $\hat{z}_j^n$ can be approximated by a noise multiplied by $z_j^s$ but applied to $\hat{z}_j^s$, if we neglect the correlation term in Eq. (14), and only the mean and variance of the noise are of interest. We can further simplify the approximation by applying multiplicative noise directly to $\hat{z}_j^s$ as

$$\hat{z}_j' = \hat{z}_j^s + \text{AsConst}\left(v_j \hat{z}_j^s\right). \tag{15}$$

The advantage of this variant is that, while Eq. (11) and Eq. (13) require an extra forward pass for each layer, Eq. (15) introduces no computational overhead, and is straightforward to implement. A similar idea was explored for fast dropout training [14].

To indicate the number of layers involved in noise generation, we refer to this variant (Eq. (15)) as NCMN-0, and the original form (Eq. (13)) as NCMN-1. We next generalize NCMN from NCMN-1 to NCMN-2, and demonstrate its connection to shake-shake regularization [5]. For convenience, we define the following two functions to indicate the pre-activations and activations of a batch-normalized layer:

$$\Psi^{l+1}\left(\mathbf{x}^l\right) = \text{BN}\left(\left(\mathbf{x}^l\right)^T \mathbf{W}^{l+1}\right), \tag{16}$$

where $\mathbf{x}^l = [x_i]$, $\mathbf{W}^{l+1} = [w_{ij}]$, $i \in \mathcal{H}^l, j \in \mathcal{H}^{l+1}$, and

$$\Phi^{l+1}\left(\mathbf{x}^l\right) = \phi\left(\gamma^{l+1} \odot \Psi^{l+1}\left(\mathbf{x}^l\right) + \beta^{l+1}\right), \tag{17}$$

where $\gamma^{l+1}$ and $\beta^{l+1}$ denote, respectively, the scaling and shifting vectors of batch normalization, $\odot$ denotes element-wise multiplication, and $\phi\left(\cdot\right)$ denotes the activation function. Let $\Psi_i^l\left(\cdot\right)$ be an element of $\Psi^l\left(\cdot\right)$, the noise component of NCMN-1 is then given by

$$\hat{z}_j^n = \Psi_j^{l+1}\left(\mathbf{u}^l \odot \mathbf{x}^l\right) - \Psi_j^{l+1}\left(\mathbf{x}^l\right), \tag{18}$$

where $\mathbf{u}^l = [u_i]$, $i \in \mathcal{H}^l$, and $j \in \mathcal{H}^{l+1}$.

We then define a natural generalization from NCMN-1 to NCMN-2 as

$$\hat{z}_k^s = \Psi_k^{l+2}\left(\Phi^{l+1}\left(\mathbf{x}^l\right)\right), \text{ and } \hat{z}_k^n = \Psi_k^{l+2}\left(\mathbf{u}^{l+1} \odot \Phi^{l+1}\left(\mathbf{u}^l \odot \mathbf{x}^l\right)\right) - \hat{z}_k^s, \tag{19}$$

and

$$\hat{z}_k' = \hat{z}_k^s + \text{AsConst}\left(\hat{z}_k^n\right), \tag{20}$$

where $k \in \mathcal{H}^{l+2}$. Different from NCMN-0 and NCMN-1, which can be applied to every layer, NCMN-2 can only be applied once every two layers. For residual networks (ResNets) in particular, NCMN-2 should be aligned with residual blocks, such that $\gamma^{l+2}\hat{z}_k' + \beta^{l+2}$ or $\phi\left(\gamma^{l+2}\hat{z}_k' + \beta^{l+2}\right)$ is the residual of a residual block, depending on which variant is used [15].

Interestingly, we can formulate shake-shake regularization in a similar way to NCMN-2. Shake-shake regularization is a regularization technique developed specifically for ResNets with two residual branches, as opposed to only one of standard ResNets. It works by averaging the outputs of two residual branches with random weights. For the forward pass at training time, one of the weights, $\alpha_1$, is sampled uniformly from $[0, 1]$, while the other one is set to $\alpha_2 = 1 - \alpha_1$. For the backward pass, the weights are either sampled in the same way as the forward pass, or set to the same value, i.e, $\alpha_1' = \alpha_2' = 1/2$. We first consider the latter case. Let $p \in \{1, 2\}$ denote the index of residual branches, then we have the signal component as

$$\hat{z}_k^s = \left(\hat{z}_{1,k} + \hat{z}_{2,k}\right)/2, \text{ where } \hat{z}_{p,k} = \Psi_{p,k}^{l+2}\left(\Phi_p^{l+1}\left(\mathbf{x}^l\right)\right), \tag{21}$$

and the noise component as

$$\hat{z}_k^n = v\left(\hat{z}_{1,k} - \hat{z}_{2,k}\right), \tag{22}$$

where $v = \alpha_1 - 1/2$ is uniformly sampled from $[-1/2, 1/2]$. Accordingly, the noisy pre-activation is also given by Eq. (20).

Similar to Eq. (5), we can define the SNR of $\hat{z}_k'$ as

$$\text{SNR}\left(\hat{z}_k'\right) = \frac{\mathbb{E}\left[(\hat{z}_k^s)^2\right]}{\mathbb{E}\left[(\hat{z}_k^n)^2\right]} = 3\left[1 + \frac{4\,\mathbb{E}\left[\hat{z}_{1,k}\hat{z}_{2,k}\right]}{\mathbb{E}\left[(\hat{z}_{1,k} - \hat{z}_{2,k})^2\right]}\right]. \tag{23}$$

Apparently, maximizing $\text{SNR}\left(\hat{z}_k'\right)$ does not affect the correlation between features from the same branch. However, if we keep the gradient of the noise component, or, equivalently, let $\alpha_1' = \alpha_1$ and $\alpha_2' = \alpha_2$, then maximizing $\text{SNR}\left(\hat{z}_k'\right)$ will encourage large $\mathbb{E}\left[\hat{z}_{1,k}\hat{z}_{2,k}\right]$ and small $\mathbb{E}\left[(\hat{z}_{1,k} - \hat{z}_{2,k})^2\right]$,

leading to highly correlated branches. On the other hand, if we consider the noise component as a constant, then only large $\mathbb{E}\left[\hat{z}_{1,k}\hat{z}_{2,k}\right]$ will be encouraged, which results in much weaker correlation, and hence the better generalization performance observed in practice. For a similar reason to that of NCMN, the batch normalization layers before Eq. (20) are crucial for avoiding optimization difficulties.

It is worth noting that setting $\alpha_1' = \alpha_2' = 1/2$ for the backward pass leads to exactly the same gradient for $\hat{z}_{1,k}$ and $\hat{z}_{2,k}$, which can also increase the correlation between the two branches. Thus, sampling $\alpha_1'$ randomly further breaks the symmetry between the two branches, which may explain its slightly better performance on large networks.

While the noise components of NCMN-2 and shake-shake are generated in different ways, they are injected into the network in the same way (Eq. (20)). Therefore, we expect similar regularization effects from NCMN-2 and shake-shake in practice. However, while adjusting the regularization strength is difficult for shake-shake, it can be easily done for NCMN by tuning the variance of multiplicative noise. Moreover, NCMN is not restricted to ResNets with multiple residual branches, and can be applied to various architectures.

# 4 Experiments

In our preliminary experiments we found that different choices of noise distributions, including Bernoulli, Gaussian, and uniform distributions, lead to similar performance, which is consistent with the experimental results for Gaussian dropout [3]. We use uniform noise with variable standard deviation in the following experiments. For fair comparison, in contrast to previous work [5, 16], we tune the hyperparameters (e.g., learning rate, L2 weight decay, noise standard deviation) separately for different types of noise, as well as for different datasets. We use ND-Adam [17] for optimization, which is a variant of Adam [18] that has similar generalization performance to SGD, but is easier to tune due to its decoupled learning rate and weight decay hyperparameters. See the supplementary material for hyperparameter settings, and practical guidelines for tuning them.

We first empirically verify the feature correlation effect of multiplicative noise, and the decorrelation effect of NCMN. To avoid possible interactions with skip connections, we use plain CNNs rather than more sophisticated architectures for this purpose. Due to the lack of skip connections, we use a relatively shallow network in case of optimization difficulties. Specifically, we construct a CNN by removing the skip connections from a wide residual network (WRN) [16], WRN-16-10, which is a 16-layer ResNet with 10 times more filters per layer than the original. Accordingly, we refer to the modified network as CNN-16-10. We train the network with different types of noise on the CIFAR-10 and CIFAR-100 datasets [19]. For each convolutional layer except the first one, we calculate the correlation between each pair of feature maps after batch normalization, and take the average of their absolute values. The results are grouped by the size of feature maps, and are shown in Fig. 1. In addition, the corresponding test error rates (average of 3 or more runs) are shown in Table 1.

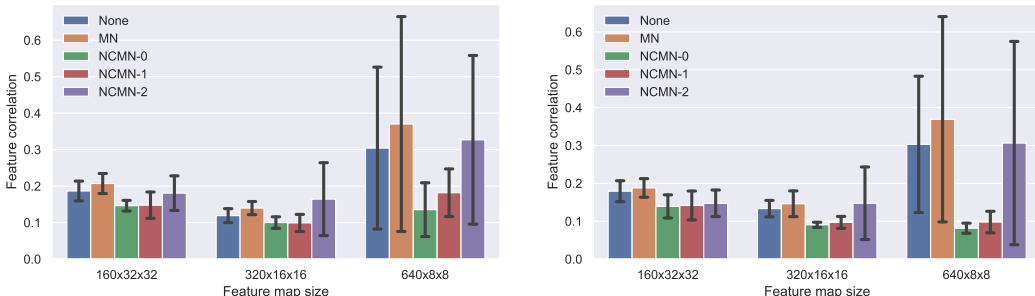

(a) Results on CIFAR-10.

(b) Results on CIFAR-100.

Figure 1: Feature correlations of CNN-16-10 networks trained with different types of noise. None refers to the baseline without noise injection, and MN refers to standard multiplicative noise. The error bars represent the standard deviation across different layers in a single run, which varies little across different runs.

Compared to the baseline, standard multiplicative noise exhibits slightly higher feature correlations for both CIFAR-10 and CIFAR-100. By modifying the gradient as Eq. (13), which results in NCMN-1, the feature correlations are significantly reduced as predicted by our analysis. Surprisingly, for all sizes of feature maps, the feature correlations of NCMN-1 reach a level even lower than that of the baseline. This intriguing result may indicate that the regularization effect of multiplicative noise strongly encourages decorrelation between features, which, however, is counteracted by the feature correlation effect. As a result, NCMN-1 significantly improves the performance of standard multiplicative noise, as shown in Table 1. As discussed in Section 3.2, NCMN-0 can be considered as an approximation to NCMN-1, which is consistent with the fact that the feature correlations and test errors of NCMN-0 are both close to that of NCMN-1. On the other hand, since NCMN-2 applies noise more sparsely (once every two layers), it exhibits higher correlations and slightly worse generalization performance than NCMN-0 and NCMN-1.

Table 1: CIFAR-10/100 error rates (%) of CNN-16-10 networks trained with different types of noise.

| Noise type | CIFAR-10 | CIFAR-100 |
|---|---|---|
| None | $4.05 \pm 0.05$ | $19.22 \pm 0.05$ |
| MN | $3.76 \pm 0.00$ | $18.08 \pm 0.03$ |
| NCMN-0 | $3.51 \pm 0.07$ | $\mathbf{17.37} \pm 0.05$ |
| NCMN-1 | $\mathbf{3.41} \pm 0.07$ | $17.55 \pm 0.06$ |
| NCMN-2 | $3.44 \pm 0.03$ | $18.16 \pm 0.04$ |

Table 2: CIFAR-10/100 error rates (%) of WRN-22-7.5 networks trained with different types of noise.

| Noise type | CIFAR-10 | CIFAR-100 |
|---|---|---|
| None | $3.68 \pm 0.02$ | $19.29 \pm 0.07$ |
| MN | $3.59 \pm 0.06$ | $18.60 \pm 0.03$ |
| NCMN-0 | $3.34 \pm 0.02$ | $17.05 \pm 0.08$ |
| NCMN-1 | $3.02 \pm 0.06$ | $17.09 \pm 0.10$ |
| NCMN-2 | $\mathbf{3.00} \pm 0.05$ | $\mathbf{16.70} \pm 0.13$ |

Next, we extend our experiments to ResNets, in order to investigate possible interactions between skip connections and NCMN. Specifically, we test different types of noise on a WRN-22-7.5 network, which has comparable performance to WRN-28-10, but is much faster to train. Since WRN is based on the full pre-activation variant of ResNets [15], NCMN-1, NCMN-2, and shake-shake require an extra batch normalization layer after each residual branch. The feature correlation is calculated for the output of each residual block, instead of for each layer.

As shown in Fig. 2, NCMN continues to lower the correlation between features, which is especially prominent in the second and third stages of the network. In addition, as shown in Table 2, we obtain even more performance gains from NCMN compared to the CNN-16-10 network. However, a notable difference is that, while NCMN-1 works well with both architectures, NCMN-0 and NCMN-2 perform slightly better, respectively with plain CNNs and ResNets, which may result from the interaction between the skip connections of ResNets and NCMN.

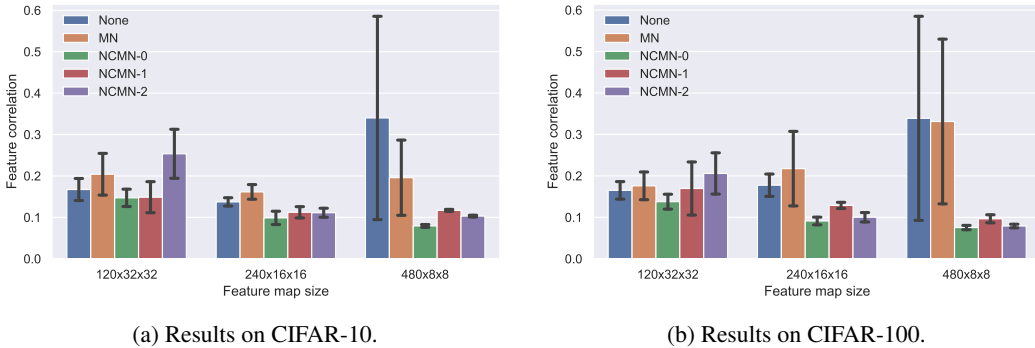

(a) Results on CIFAR-10.  (b) Results on CIFAR-100.

Figure 2: Feature correlations of WRN-22-7.5 networks trained with different types of noise.

To compare the performance of NCMN with shake-shake regularization, we train a WRN-22-5.4 network with two residual branches, which has comparable number of parameters to WRN-22-7.5. We refer to this network as WRN-22-5.4$\times$2. The averaging weights of residual branches are randomly sampled for each input image, and are independently sampled for the forward and backward passes. The training curves corresponding to different types of noise are shown in Fig. 3. On both CIFAR-

10 and CIFAR-100, NCMN and shake-shake show stronger regularization effect than standard multiplicative noise, as indicated by the difference between training (dashed lines) and testing (solid lines) accuracies. However, we make the observation that, the regularization strength of shake-shake is stronger at the early stage of training, but diminishes rapidly afterwards, while that of NCMN is more consistent throughout the training process. See Table 3 for detailed results.

As shown in Table 3, we provide additional results demonstrating the performance of NCMN on models of different sizes. Interestingly, NCMN is able to significantly improve the performance of both small and large models. It is worth noting that, apart from the difference in architecture and the number of parameters, the number of training epochs can also notably affect generalization performance [20]. Compared to the results of shake-shake regularization, a WRN-28-10 network trained with NCMN is able to achieve comparable performance in 9 times less epochs. For practical uses, NCMN-0 is simple, fast, and can be applied to any batch-normalized neural networks, while NCMN-2 yields better generalization performance on ResNets.

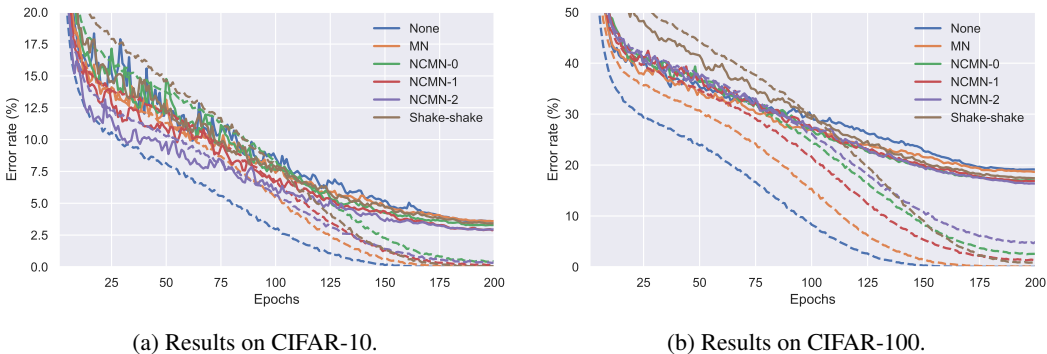

(a) Results on CIFAR-10.  (b) Results on CIFAR-100.

Figure 3: Training curves of WRN-22-7.5 networks trained with different types of noise, and that of a WRN-22-5.4×2 network trained with shake-shake regularization.

Table 3: More results on CIFAR-10/100 for comparison.

| Model | Params | Epochs | Noise type | CIFAR-10 | CIFAR-100 |
|---|---|---|---|---|---|
| DenseNet-BC $(250, 24)$ [21] | 15.3M | 300 | None | 3.62 | 17.60 |
| ResNeXt-26 $(2\times96d)$ [5] | 26.2M | 1800 | Shake/None | **2.86**/3.58 | — |
| ResNeXt-29 $(8\times64d)$ [5] | 34.4M | 1800 | Shake/None | — | **15.85**/16.34 |
| WRN-28-10 [16] | 36.5M | 200 | Dropout/None | 3.89/4.00 | 18.85/19.25 |
| DenseNet-BC $(40, 48)$ | 3.9M | 300 | NCMN-0/None | 3.51/4.07 | 17.68/19.92 |
| CNN-16-3 | 1.6M | 200 | NCMN-0/None | 4.47/5.10 | 21.92/24.97 |
| CNN-16-10 | 17.1M | 200 | NCMN-1/None | 3.41/4.05 | 17.55/19.22 |
| WRN-22-2 | 1.1M | 200 | NCMN-0/None | 4.56/5.19 | 23.54/25.90 |
| WRN-22-7.5 | 15.1M | 200 | NCMN-2/None | 3.00/3.68 | 16.70/19.29 |
| WRN-22-5.4×2 | 15.5M | 200 | Shake/None | 3.51/4.04 | 17.77/19.71 |
| WRN-28-10 | 36.5M | 200 | NCMN-2/None | **2.78**/3.70 | **15.86**/18.42 |

We also experimented with language models based on long short-term memories (LSTMs) [22]. Intriguingly, we found that the hidden states of LSTMs had a consistently low level of correlation (less than $0.1$ on average), even in the presence of dropout. Consequently, we did not observe significant improvement by replacing dropout with NCMN.

## 5 Conclusion

In this work, we analyzed multiplicative noise from an information perspective. Our analysis suggests a side effect of dropout and other types of multiplicative noise, which increases the correlation between features, and consequently degrades generalization performance. The same theoretical framework also provides a principled explanation for the performance gain of shake-shake regularization.

Furthermore, we proposed a simple modification to the gradient of noise components, which, when combined with batch normalization, is able to effectively remove the feature correlation effect. The resulting method, NCMN, outperforms standard multiplicative noise by a large margin, proving it to be a better alternative for batch-normalized networks.

While we combine batch normalization with NCMN to counteract the tendency of increasing feature magnitude, an interesting future work would be to investigate if other normalization schemes, such as layer normalization [23] and group normalization [24], can serve the same purpose.

## Acknowledgments

This work was supported by NSFC 61628209, Hubei Science Foundation 2016CFA030, 2017AAA125, and Wuhan Science & Tech Program 2018010401011288.

## Footnotes

[1]Code is available at `https://github.com/zj10/NCMN`.

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
