[Supplementary Material · supplementary.pdf]

# Supplementary Material for Removing the Feature Correlation Effect of Multiplicative Noise

**Zijun Zhang**
University of Calgary
zijun.zhang@ucalgary.ca

**Yining Zhang**
University of Calgary
yining.zhang1@ucalgary.ca

**Zongpeng Li**
Wuhan University
zongpeng@whu.edu.cn

## A  Hyperparameter Settings and Practical Guidelines

The hyperparameter settings for the experiments are listed in Table 1. We use ND-Adam for most of the experiments, except for DenseNet, where stochastic gradient descent (SGD) with a momentum of 0.9 is used. A cosine learning rate schedule (monotonically decreasing) is used for all experiments. It is worth noting that the optimal value of weight decay (applied to biases for ND-Adam, and to both weight vectors and biases for SGD) varies in different cases. However, it is shown that when using SGD, the *effective learning rate* is coupled with both the learning rate and the weight decay hyperparameters [1]. Consequently, to tune weight decay without affecting the effective learning rate, one needs to change the learning rate hyperparameter inversely proportional to the value of weight decay, as exemplified by the settings for DenseNet.

Table 1: Hyperparameter settings. $\sigma$, $\alpha_0$, and $\lambda$ are, respectively, the noise standard deviation, the initial learning rate, and the weight decay factor.

| Model | Noise type | $\sigma$ (C10/C100) | $\alpha_0$ (C10/C100) | $\lambda$ (C10/C100) |
|---|---|---|---|---|
| CNN-16-3 | None | — | 0.04 | 5e−6/1e−3 |
| | NCMN-0 | 0.15/0.1 | 0.04 | 5e−6/5e−5 |
| CNN-16-10 | None | — | 0.04 | 5e−6/1e−3 |
| | MN | 0.35/0.25 | 0.04 | 5e−5/1e−3 |
| | NCMN-0 | 0.35/0.25 | 0.04 | 5e−6/2e−5 |
| | NCMN-1 | 0.35/0.25 | 0.04 | 5e−5/1e−3 |
| | NCMN-2 | 0.35/0.25 | 0.03/0.04 | 2e−5/1e−3 |
| WRN-22-2 | None | — | 0.04 | 5e−6/1e−3 |
| | NCMN-0 | 0.15/0.1 | 0.04 | 5e−6/5e−5 |
| WRN-22-7.5 | None | — | 0.04 | 5e−6 |
| | MN | 0.35/0.25 | 0.04 | 5e−6/2e−5 |
| | NCMN-0 | 0.35/0.25 | 0.04 | 5e−6/2e−5 |
| | NCMN-1 | 0.35/0.25 | 0.04 | 5e−6/2e−4 |
| | NCMN-2 | 0.4/0.3 | 0.03/0.04 | 2e−5/2e−4 |
| WRN-22-5.4×2 | None | — | 0.04 | 5e−6 |
| | Shake | — | 0.04 | 5e−6/2e−4 |
| WRN-28-10 | None | — | 0.04 | 5e−6 |
| | NCMN-2 | 0.45/0.35 | 0.03/0.04 | 2e−5/2e−4 |
| DenseNet-BC (40, 48) | None | — | 0.1 | 2e−4 |
| | NCMN-0 | 0.2 | 0.05 | 4e−4 |

To investigate the sensitivity of generalization performance to the noise level of NCMN, as well as the relationship between network width and optimal noise level, we train multiple WRNs of different widths with NCMN-0. We set $\alpha_0 = 0.04$, $\lambda = 2\mathrm{e}{-5}$, and vary the noise variance, $\sigma^2$. As shown in Fig. 1, the test error rate drops more significantly when $\sigma^2$ is small. More importantly, it indicates a roughly linear relationship between network width and optimal noise variance, which can be useful for determining the value of $\sigma$.

Figure 1: CIFAR-100 error rates of WRNs with different widths and noise variances. Minima are marked by plus signs.

# References

[1] Zijun Zhang, Lin Ma, Zongpeng Li, and Chuan Wu. Normalized direction-preserving adam. *arXiv preprint arXiv:1709.04546*, 2017.