[Reviews · NeurIPS 2018]

Reviewer 1



Summary: This paper presents an analysis of multiplicative noise based regularizers (e.g. DropOut) and demonstrates that employing multiplicative noise leads to internal features becoming correlated, as the regularized objective favors increasing the signal-to-noise ratio by adding redundancy. The authors propose to decompose the pre-activations of the noised layers into their signal and noise components (which is tractable since the noise under consideration is artificial) and remove the effects of the noise from the backward pass. They show that this removal changes the regularized objective to not encourage feature correlation (although it does not, as far as I can tell, penalize it, either). A connection is drawn between the proposed regularizer and shake-shake (another backward-pass-hacking multiplicative-noise based regularizer), providing (for the first time I’m aware of) some analytical justification for the latter. The results are justified through standard benchmarks on CIFAR, where they demonstrate competitive (potentially even SOTA) results for their method. The paper is well-motivated, well-written, clear, and easy to follow. The technique proposed is simple and should be easy to implement and reproduce, without requiring more than small architectural changes. I found the analytical results convincing, and the connection to shake-shake (which, along with its variants, has thus far been an ad hoc hack) worthwhile. I found the empirical results as acceptable (the scores on CIFAR are certainly good) but lacking error bars on test error (which I don’t believe we should be letting authors get away with) or results on larger-scale tasks, particularly ImageNet (see my comments below). I think this is worth the attention of the NIPS community and vote in favor of acceptance. General notes: -The insight that features can be decomposed into the noise and signal (because the noise is artificial) is intriguing and clever. I found equation 15 particularly elegant. -Please re-run your CIFAR experiments and report error bars. While common practice may be to *not* do this, I don’t think reviewers should be letting authors get away with this in situations where it should obviously be done. Even on a modest GPU these experiments can be run in under a day, so it should be no challenge to at least take the average of 3 and report variances for at least the top-scoring run. -My primary complaint with this paper and with the shake-shake line of research is that no one, to my knowledge, has demonstrated that these techniques work at a scale beyond CIFAR. CIFAR-10 in particular has 4 times as many images per class as ImageNet, and it's known to be a regularization game, but for other tasks which are perhaps more practical, it’s not clear that these techniques provide any benefit. I don’t *fault* the authors for not running ImageNet experiments as it would be unreasonable to expect every group to test on large scale tasks, but I suspect that if the authors want to convince the broader community of the value of their work, they could easily do so by being the first to demonstrate that these techniques help on ImageNet (or that they don’t—such a negative result would also improve this paper). -Calling it Feature-Decorrelating multiplicative noise may not be quite correct—as far as I can tell (and the authors are welcome to correct me on this front), nothing in the regularized objective actually encourages decorrelation between features. A more accurate name might be something along the lines of "Non-Correlating Multiplicative Noise." While it appears that in practice variants 0 and 1 have lower feature correlations than the un-noised baseline, variant 2 often has slightly higher feature correlations (though lower than the MN baseline). This is not discussed anywhere in the paper that I can find, and I think it should be. I’m curious as to the authors’ opinion on this anyhow—is it because the noise is only applied once every two layers, or what? -What exactly are the error bars in the bar graphs of e.g. figure 1? Are they representing the variance of feature correlation across runs with multiple different settings, or across different layers at a given resolution in a single run? Please state this explicitly in the paper. -I found the description of hyperparameters confusing. The statements “the learning rate is set to either 0.03 or 0.04” implies that either different settings are used for different experiments, or grid search was used and the results were then averaged, but this is not clarified anywhere. If the authors wish to move hyperparameter details into supplementary material to save space in the body of the paper, that’s fine, but the way it is currently written is vague and might hinder reproduction. -The authors might consider leveraging empirical successes of MN strategies as an auxiliary justification for using batchnorm—currently batchnorm is often used because “it just works,” but if one finds that FDMN is useful but only works when batchnorm is in the mix, then one might justify batchnorm as “I need FDMN and it only works with batchnorm, a combination which is justified by the analysis in the FDMN paper.” -I briefly tried coding up FDMN-0 several different ways for WRN-40-4 and WRN-28-10 but didn’t really see a change in performance on CIFAR-100+. I didn't try very hard, so there could of course be mistakes in my implementation, and I have not factored this into my score . One implementation detail that was not clear to me was how to apply this to pre-activation style ResNets—should the noise be added at the batchnorm’d outputs before nonlinearities, without any other changes, or should the batchnorm orders be changed (an additional BN is mentioned for variants 1 and 2)? -My implementation also made me wonder if the authors truly mean that they set the standard deviation of their uniform noise to 0.25 for CIFAR-100—this results in uniform noise U(-sqrt(3)/2, sqrt(3)/2), which seems like a very large noise component. -Minor: -line 19, “However, such over-parameterized models are more prone to overfitting.” It’s not presently clear that this is actually true; the existence of a generalization gap does not necessarily imply overfitting, and models often continue to improve their validation score even as they squeeze the training loss ever closer to zero. -line 132: I found the use of parentheses in “mean (variance)” momentarily confusing. I take it to mean “mean and variance.” -The metric chosen for feature correlation seems reasonable, though in an ideal world it would be good to try methods particularly targeted at analyzing the similarity of internal representations in deep nets, e.g. SVCCA [NIPS 17]. The authors needn’t feel obliged to do this, though. Response to author feedback: I feel that the authors' response has adequately addressed my concerns and my inclination is to score this paper as more of an 8 than a 7. I think that the line of work on "gradient-hacking" regularizers like Shake-Shake and ShakeDrop is interesting and potentially very valuable (the techniques show a lot of promise). Recent papers in this line of research have been rejected from the mainstream conferences, perhaps because the papers themselves were somewhat lacking. This method introduces fewer design choices (the choice of distribution and the variance of the noise) than the Shake* regularizers, which have myriad moving parts and necessitate architectural changes. As a practitioner I would much rather drop this noise into my code. To me this paper does a good job of justifying this style of technique by properly connecting it to better-understood existing methods, provides mathematical detail but doesn't fall prey to "mathiness", has good, although not quite "perfect," experimental results, and (apparently) has open source code. I argue more strongly in favor of acceptance.

Reviewer 2



Summary: The authors study multiplicative noise using theoretical tools as well as numerical experiments. They demonstrate how feature correlation (arising from use of naive multiplicative noise) degrades model performance, and develop a regularizer which boosts performance by removing the correlation generated by these methods. The authors relate this new regularizer to shake-shake normalization, theoretically. Finally, they support their findings with considerable experimental work. Strengths: The paper is clear, compelling, and reasonably well supported by the theoretical and numerical arguments presented within. The effort to connect the work theoretically to shake-shake regularization is particularly useful and intuitively helpful. The work is clearly original, and could be useful in the community. Weaknesses: It was not obvious how sensitive the method is to hyper parameter tuning (i.e., sensitivity to standard deviation of noise used). For example, the authors mention (line 37): “Moreover, these techniques require extra hyperparameters to control the strength of the penalty, 38 which further hinders their practical application.” But then, on line 226: “As a side note, we empirically find that the optimal noise 226 variance tends to be proportional to the width of the network, if other settings remain the same.” I think this is a pretty important plot to show! I am convinced that the *type* of noise doesn’t change the results too much, but seeing a simple plot of test performance vs. standard deviation of noise scale would solidify the experimental work section. Overall, it was a solid contribution, worthy of acceptance.

Reviewer 3



Quality: The work focuses on feature correlation effect of dropout in deep neural networks (DNN).The authors propose a method to reduce the increasing feature correlation effect that occurs as one tries to increase the signal-to-noise ratio (SNR) of representations. The propose feature decorrelation techniques exploit the batch normalization which effectively controls the tendency of increasing feature magnitude. The work seems to connect well with the existing literature and provide a effective way of improving the performance of multiplicative noise in image classification tasks. Clarity: The objective of the problem and the theoretical results by and large have been presented in a clear way. However, I didn't quite understand the unified view in section 3.2 that presents the multiple variants of the FDMN-0, FDMN-1 etc. I agree that the authors presented well in the previous section how FDMN could be used to effectively reduce the feature correlation effect using batch normalization. But the presentation of the different variants of FDMN was not quite clear to me. More specifically, the equation representing FDMN-0 in (15) and how the noise components are generated in FDMN -2 in equation (19) may need some more remarks or discussions. Novelty: I feel the work has potential and the authors present a simple but effective approach of reducing the feature correlation in dropout. Certainly, it's a novel contribution to the existing literature of DNN. However, I feel the proposed technique could have been presented with a bit more clarity to show how it is more effective than the existing regularization techniques in DNN. Significance: This area of work is significant to the literature discussing regularization in DNNs. I have my reservation with the multiple variants of FDMN. I am not entirely convinced how the noise component is generated in those different versions. Hence, I can't judge the actual importance of the current techniques in comparison to the existing regularization methods. --------------------------------------------------------------------------------------- I read the author response. I am happy with the author feedback for my queries. I think they have done well enough in justifying the motivations behind different variants of FDMN (which was my primary concern). Further, the comparison with shake-shake regularizer has been explained well therein. Overall, I am happy with the detailed response of the author(s).